# Mechanisms of RNA and Protein Quality Control and Their Roles in Cellular Senescence and Age-Related Diseases

**DOI:** 10.3390/cells11244062

**Published:** 2022-12-15

**Authors:** Donghee Kang, Yurim Baek, Jae-Seon Lee

**Affiliations:** 1Research Center for Controlling Intercellular Communication (RCIC), College of Medicine, Inha University, Incheon 22212, Republic of Korea; 2Department of Molecular Medicine, College of Medicine, Inha University, Incheon 22212, Republic of Korea; 3Program in Biomedical Science & Engineering, Inha University, Incheon 22212, Republic of Korea

**Keywords:** RNA quality control, protein quality control, cellular senescence, age-related diseases

## Abstract

Cellular senescence, a hallmark of aging, is defined as irreversible cell cycle arrest in response to various stimuli. It plays both beneficial and detrimental roles in cellular homeostasis and diseases. Quality control (QC) is important for the proper maintenance of cellular homeostasis. The QC machineries regulate the integrity of RNA and protein by repairing or degrading them, and are dysregulated during cellular senescence. QC dysfunction also contributes to multiple age-related diseases, including cancers and neurodegenerative, muscle, and cardiovascular diseases. In this review, we describe the characters of cellular senescence, discuss the major mechanisms of RNA and protein QC in cellular senescence and aging, and comprehensively describe the involvement of these QC machineries in age-related diseases. There are many open questions regarding RNA and protein QC in cellular senescence and aging. We believe that a better understanding of these topics could propel the development of new strategies for addressing age-related diseases.

## 1. Cellular Senescence: Types and Characteristics

Cellular senescence is a hallmark of aging that is characterized by stable cell cycle arrest in response to various cellular damages and stresses [1]. Cellular senescence was first described in 1961 by Hayflick and Moorhead, who observed that a human diploid fibroblast cell line ceased to proliferate after an extended number of population doublings [2]. This process, termed replicative senescence, was later related to telomere shortening [3]. Cellular senescence can also be induced by various intrinsic and extrinsic stressors, such as DNA damage, epigenetic alterations, oxidative stress, oncogene- and therapy-induced stress, inactivation of tumor suppression, and viral infections [4,5,6,7,8]. Senescent cells have morphological and molecular features that differ from those of normal cells, including increased SA-β-gal activity at pH 6 (the intra-lysosomal pH), the presence of lysosomal granules in the cytoplasm, and flattened and enlarged morphologies [9]. Cellular senescence is mainly initiated by two tumor suppressor pathways: the p53-p21 and p16 ^Ink4A^ -RB pathways. Upon p53 activation, the up-regulation of p21 (also known as CDKN1A) blocks the formation of cyclin-cyclin-dependent kinase (CDK) complexes. p16^Ink4A^ (also known as CDKN2A) directly binds to and inhibits the interaction between CDK4/6 and cyclin D. When CDKs are inhibited by p21 and p16^Ink4A^, RB remains hypophosphorylated and interacts with the transcription factor, E2F, leading to irreversible cell cycle arrest [1,10]. Although senescent cells are characterized by growth arrest, they remain metabolically active. They secrete not only extracellular vesicles, but also various bioactive proteins, termed senescence-associated secretory phenotypes (SASPs), which include inflammatory cytokines, chemokines, matrix metalloproteinases (MMPs), growth factors [11]. SASPs reinforce senescence themselves as autocrine messengers. In paracrine manner, SASPs regulate the proliferation, angiogenesis, invasion, and metastasis of tumors, local immune responses, and the cellular reprogramming of surrounding cells [12,13]. Cellular senescence plays beneficial roles in tumor suppression, immune cell recruitment, wound healing, fibrosis reduction, and embryo development. However, senescent cells also play detrimental roles in the proliferation, angiogenesis, tumor metastasis, tumor invasiveness, chronic inflammation, fibrosis, and tissue regeneration. Cellular senescence is closely associated with multiple chronic diseases, such as cancers and neurodegenerative, muscle, and cardiovascular diseases [14].

## 2. QC of RNA

In eukaryotic cells, RNA biogenesis comprises multiple processes involving transcription, capping, splicing, 3′ end cleavage, polyadenylation, nucleus-cytoplasm export, and translation. During these processes, abnormal or nonfunctional RNAs are degraded by RNA QC mechanisms to avoid the production of dysfunctional or toxic proteins [15]. Below, we describe the mechanisms of RNA QC in the nucleus and cytoplasm and their implications during cellular senescence and aging (Figure 1).

### 2.1. QC of RNA in the Nucleus

In the cell nucleus, pre-mRNAs are synthesized from genes by RNA polymerase II (RNA Pol II) and then processed into mRNAs via splicing, 5′ end capping, 3′ end endonucleolytic cleavage, and polyadenylation. Each generated mRNA is then packed into a messenger ribonucleoprotein particle (mRNP) by the binding of RNA-binding proteins (RBPs), and exported to the cytoplasm [16]. During the various RNA processing events, numerous RNA QC systems target abnormal nuclear RNAs, retaining and degrading them within the nucleus or exporting them to the cytoplasm for degradation.

Incomplete mRNAs are retained within the nucleus by fail-safe mechanisms, as they possess either retention features (retained and detained introns, or extended poly (A) tails) or cis-acting elements (GUAUGUU, the pentamer AGCCC, or polypyrimidine tract) for the interaction of retention factors [17]. Retained introns (RIs) have two types, such as ‘poison-cassette’ exon and ‘detained’ introns. ‘Poison-cassette’ exon-contained mRNAs have premature termination codon (PTC) in their exon, are exported to the cytoplasm, and are degraded by nonsense-mediated RNA decay (NMD). Second type of RI is ‘detained’ introns (DI). mRNAs may frequently have the detained introns (DIs), which have weaker 5′ and 3′ splice sites than spliced introns. DIs interact with inactive pre-spliceosome (including U2AF, UAP56, and SF3B1) and SR proteins, and detained intron-containing RNAs are thereby localized to nuclear speckles [18]. Pre-mRNA or mRNA, which interacts with retention factors or lacks export factors, are retained in the nucleus and ultimately degraded by nuclear exosomes [19]. A nuclear exosome embodies two different forms of nuclear RNA degradation, namely those driven by an 11-subunit complex containing EXOSC10/RRP6 and DIS3, and those driven by a nucleolar 10-subunit complex that contains EXOSC10/RRP6 but lacks DIS3. Nuclear co-factors, such as MPP6 and C1D, bind to the exosome to recruit MTR1, which is a helicase of the SKI2-like family. TRAMP is an exosome accessory complex that consists of non-canonical poly(A) polymerase, hTRF4-2 (PAPD5), ZCCHC7, and MTR4 helicase. TRAMP prepares aberrant RNAs for degradation by adding a poly(A) tail at the 3′ end of RNA substrates (including pre-rRNAs, abnormal tRNAs, snRNAs, snoRNAs, and ncRNAs) [20,21,22].

### 2.2. QC of RNA in the Cytoplasm

QC mechanisms in the cytoplasm degrade aberrant mRNAs that could potentially be translated into toxic proteins. Adaptor proteins bind to the aberrant mRNAs and promote their degradation via various degradation pathways [23]. NMD is a highly conserved mRNA-decay pathway that selectively degrades mRNAs containing PTC or NMD-activating characters. NMD regulates both the QC and the quantity control of mRNAs. In NMD, UPF1 serves as a central regulator that recruits degradation enzymes, including SMG6 (or SMB5-SMG7), XRN1, CCR4-NOT, DCP2, and PNRC2 [24]. NMD can target and degrade mRNAs through two mechanisms: 3′ untranslated region (3′ UTR) EJC-dependent and EJC-independent NMD. The exon junction complex (EJC) is one of protein complexes which plays a crucial role in post-transcriptional regulation for gene expression. The EJCs serve to mark exon-exon junctions after the intron removal and are displaced by ribosomes during translation [25]. In the 3′ UTR EJC-dependent mechanism, the aberrant mRNAs have a PTC generated by biosynthesis defects or nonsense mutation. The PTC is generally located ~50–55 nucleotides upstream of an exon-exon junction. During the splicing process, an EJC composed of eIF4A3, RBM8A/Y14, and MAGOH, binds ~20–24 upstream nucleotides of upstream of the mRNA exon-exon junctions. RNPS1 and UPF3X/UPF3B join the EJC, and then UPF2 binds to UPF3X/UPF3B. UPF1 interacts with UPF2 and UPF3X/UPF3B and undergoes a conformational change to stimulate its ATPase and helicase activities. The SMG1-SMG8-SMG9 termination complex binds to UPF1 and induces the SMG1-mediated phosphorylation of UPF1. Next, phosphorylated UPF1 interacts with SMG5-SMG7 or SMG6 to mediate exonucleolytic or endonucleolytic mRNA decay, respectively. Abnormal mRNAs with long unstructured 3′ UTRs can be degraded by EJC-independent NMD. In this process, UPF1 binds to SMG1-SMG8-SMG9 and is phosphorylated independent of EJC. Translation is repressed and decay factors are recruited to target mRNAs, leading to their degradation [24,26].

Aberrant mRNAs can be degraded by other RNA QC mechanisms, such as nonstop decay (NSD), no-go decay (NGD), rapid tRNA decay (RTD), nonfunctional rRNA decay (NRD), and ribosome extension-mediated decay (REMD) [27,28,29,30,31,32]. NSD is an RNA surveillance pathway that acts to decay mRNAs lacking stop codons. NAD requires the recruitment of Ski7 (eEF1A) and the Dom34–Hbs1 complex, and induces the 3′-5′ degradation of a target mRNA [27,28,29]. NGD targets defective mRNAs that exhibit stalled elongation, wherein the ribosome is stalled and the A site of ribosome is empty. The Dom34/Hbs1 complex associates with the A site of ribosome and mediates the hydrolysis of peptidyl-tRNA; this leads to release of the peptide or peptidyl-tRNA, which is then degraded by ubiquitin-mediated proteasomal degradation. Defective mRNA is cleaved by an endonuclease and the stalled ribosome is released from the mRNA. Next, the mRNA fragments are exonucleolytically digested by Xrn1 and the exosome [30]. RTD degrades tRNAs lacking several body modifications; here, the tRNA is processed by the 5′-3′ exonucleases, Rat1 and Xrn1 [32]. REMD is a cell type-specific mRNA decay pathway that induces a poly(A)-dependent destabilization of mRNAs [31].

## 3. QC of Proteins: Proteostasis

Proteostasis is enacted by a complex protein QC network that monitors protein translation, folding, aggregation, localization, and degradation. Proteostasis maintains protein homeostasis through several main machineries, including chaperone proteins, the ubiquitin-proteasome system, and autophagy (Figure 2).

### 3.1. Ubiquitin-Proteasome System (UPS)

The proteasome is a macromolecular complex that is responsible for degrading abnormal, damaged, and unnecessary proteins through ubiquitin-dependent or -independent pathways. The major proteolytic mechanism of protein degradation is the ubiquitin-proteasome system (UPS), which requires that the target protein be tagged with a 76 amino-acid peptide called ubiquitin (Ub) [33]. The UPS maintains the proper concentrations of regulatory proteins involved in various biological processes, including apoptosis, cell cycle progression, proliferation, differentiation, angiogenesis, and cellular senescence [34,35].

The UPS specifically controls target proteins (substrates) via enzymatic procedures that involve the E1 ubiquitin-activating enzyme, E2 ubiquitin-conjugating enzyme, and E3 ubiquitin ligase [36]. Ub is first activated by E1 in an ATP-dependent manner: E1 hydrolyzes ATP and adenylates the glycine residue at the carboxyl-terminus of Ub. Then, the activated Ub is transferred and linked to a cysteine site of E2. In the final step, E3 attaches Ub to the ε-amino group of lysine residues of the substrate protein [37,38]. E3 binds to the primary sequence motif of substrates, and thereby provides specificity to the UPS [39]. Ub sequentially conjugates to the primary ubiquitin of substrates and forms a polyubiquitin (Poly-Ub) chain acts as a signal for the substrate protein to be degraded by the proteasome [38].

The proteasome consists of two major assemblies: the 19S regulatory particle (RP) and the 20S core particle (CP). The RP has two catalytic activities, ATP hydrolysis and proteolytic cleavage, and consists of 19 different subunits that form two heteromeric subcomplexes, called the lid and base complexes. The base consists of six different ATPases called regulatory particle triple-A proteins 1 (RPT1)-RPT6, and three non-ATPase subunits called regulatory particle non-ATPase1 (RPN1), RPN2, and RPN13. The ATPase subunits are involved in substrate unfolding and α-ring channel opening. RPN1, RPN13, RPT5, and RPN10 act as ubiquitin receptors, capturing ubiquitylated proteins. RPN10 forms a link between the base and lid. The lid, which is composed of nine non-ATPase subunits (RPN3, RPN5–RPN9, RPN11, RPN12, and RPN15), deubiquitinates the captured substrates to facilitate their degradations. The other lid subunits have yet to be studied in detail. Thus, RP recognizes ubiquitin tags, then deubiquitinates, unfolds, and translocates the substrate into the 20S core particle, and finally degrades the substrate into peptides [40,41,42]. Substrate degradation occurs inside the CP, which is a barrel-shaped cylinder composed of α- and β-rings that form four heteroheptameric rings. The two outer α-rings (α1–α7) function as a gate that associates with regulatory particles. The two inner rings are formed by β subunits (β1–β7). β1 has caspase activity, β2 has trypsin-like activity, and β5 possesses chymotrypsin-like activity. Other β subunits (β3, β4, β6, and β7) form an inner ring structure [41,42,43].

### 3.2. Autophagy

Autophagy is enacted by a conserved lysosomal pathway that degrades cytoplasmic components and organelles. This pathway can be induced by diverse stresses, including the deprivation of nutrients or growth factors or the presence of reactive oxygen species, defective organelles, misfolded proteins, DNA damage, or intracellular pathogens [44,45]. The UPS degrades short-lived, unfolded, and misfolded proteins, while autophagy targets long-lived proteins and organelles, such as the endoplasmic reticulum, mitochondria, and peroxisomes. The UPS and autophagy can engage in cross-talk and affect one other [46]. There are three major forms of autophagy: macroautophagy, microautophagy, and chaperon-mediated autophagy (CMA).

Macroautophagy, which is hereafter referred to as autophagy, sequesters cytosolic cargos within double-membrane vesicles called autophagosomes, and subsequently delivers them to lysosomes for degradation. This process can be nonselective (bulk) or selective. Nonselective autophagy randomly targets subcellular organelles or macromolecules in bulk, whereas selective autophagy specifically recognizes and degrades cargos through autophagy receptors. There are various forms of selective autophagy, including aggrephagy (acting on aggregated proteins), glycophagy (glycogen), lipophagy (lipid droplets), mitophagy (mitochondria), nucleophagy (nuclear materials), and xenophagy (pathogens) [47,48]. Under various stress conditions, autophagy is induced by a multistep process that includes initiation, membrane nucleation, phagophore formation, phagophore expansion, lysosome fusion, and degradation. Autophagy is tightly regulated by the inhibitor, mTOR, and the activator, AMP-activated kinase (AMPK). UNC51-like kinase 1 (ULK1) is a key initiator of autophagy; it is phosphorylated and activated by dissociation of mTOR or interaction with AMPK. Additional crucial autophagy proteins in mammals include the light chain 3 (LC3)/γ-aminobutyric acid receptor-associated proteins (GABARAPs). LC3 is first processed by a member of the autophagy-related protein 4 (ATG4) family to form cytosolic LC3-I. Next, LC3-I is translocated to a phagophore through ATG7 and ATG3, and becomes conjugated to phosphatidylethanolamine (PE) to form LC3-II (lipidated LC3). LC3-II interacts with specific autophagy receptors (SARs) or cargo receptors harboring LC3-inteacting motifs (LIRs) [48,49]. Autophagy mediated via the recognition of specific autophagy receptors is well established as selective autophagy, although it was originally known as non-selective autophagy [50,51].

Microautophagy is the directly uptake of cytosolic cargos by membrane protrusion and either invagination of the lysosomal or late endosomal membranes. This process can be non-selective or selective. Non-selective microautophagy has two different types, fission-type and fusion-type. Selective microautophgy takes specific cargos depending on the cellular context, including micromitophagy, microreticulophagy, micronucleophagy, microlysophagy, macrolipophagy, microproteophagy, and endosomal microautophagy (eMI) [52].

Chaperon-mediated autophagy (CMA) drives the selective uptake of KFERQ-like motif-harboring proteins into lysosomes. A cytosolic cargo with a KFERQ-like motif is recognized by Hsc70 and co-chaperons. The cargo-chaperon complex binds to lysosome-associated membrane protein type 2A (LAMP2A), and the cargo is unfolded by the chaperon complex and forms the CMA translocation complex. The cargo is translocated by lysosomal Hsc70 (Lys-Hsc70) and undergoes degradation by lysosomal proteases, such as cathepsin A and MMPs. Finally, LAMP2A is dissociated from the translocation complex [47].

### 3.3. QC of RNAs and Proteins in Cellular Senescence and Aging

Several studies have reported that NMD activity is closely associated with aging: It is reduced during aging processes, resulting in the accumulation of abnormal RNAs due to failure of RNA QC. For example, NMD activity was shown to be decreased in various tissues of aged C. elegans and required for the longevity of this model organism [53]. Several senescent cell models induced by various stimuli show accumulation of exosome substrate RNAs with long promoters, which indicates reductions in NMD activity and the expression of exosome subunits, including EXOSC2, EXOSC3, EXOSC6, EXOSC8, EXOSC9, EXOSC10 and DIS3L [54].

Proteostasis provides a balanced cellular proteome, wherein the cycles of protein synthesis and degradation are fine-tuned via UPS and autophagy. Loss of proteostasis has been observed in various aged human tissues and organs, such as skin, muscle, lymphocytes, liver, lung, heart, kidney, and brain, and is considered a hallmark of cellular senescence and aging [55,56,57,58,59,60,61]. The loss of proteostasis increases the levels of damaged and dysfunctional proteins, which eventually form large toxic protein aggregates. The accumulation of abnormal proteins is frequently found in age-related diseases, suggesting that proteostasis loss is a strong contributor to age-related diseases [62,63].

An age-dependent decrease of UPS can occur at different levels, such as reduced proteasomal activity, structural alterations and/or subunit replacement of proteasomes, decreased expression of proteasomal subunits, and proteasomal modification [38,64]. The activity levels of the 20S and 26S proteasomes are decreased in aged liver, brain, and muscle when compared with their young counterparts [65,66,67]. The relative attachment of PA28 and PA700 to the 20S core proteasome is lower in aged muscle than in young muscle [68]. The expression levels of proteasomal subunits are decreased in old human skin cells or mouse liver tissues [69,70]. The expression levels of PSMA1 (α subunit), PSMB6, PSMB5, PSMB4 (β subunits), PSMC4, PSMD8 (19S regulator subunits), PSME1, and PSME2 (11S activators) are lower in skin cells from elderly donors compared to young donors [69]. In liver tissues from old mice, PSMD4 expression is reduced [70]. Moreover, stress-induced modification of proteasomal subunits, such as 4-hydroxynonenal, carbonylation, S-glutathionylation, poly ADP-ribosylation, S-nitrosylation, phosphorylation, or ubiquitination, can reduce proteasomal activity during cellular senescence and aging [71,72,73,74,75].

Alterations of the UPS and/or dysfunction of autophagy and CMA are found during cellular senescence and aging. The dysfunctions of autophagy are heterogeneous. At the transcriptional level, key autophagy genes, such as ATG5, ATG7, and BECN1, are downregulated in human aged brain tissues [76]. Histone modification also regulates the expression of autophagy-related genes. For instance, autophagy is induced by a decrease in H4K16 acetylation (H4K16Ac) and/or an increase in Acetyl-CoA or H3K9 demethylation (H3K9me2) [77]. Reduction of the histone acetyltransferase, hMOF, decreases H4K16Ac in the promoter regions of ULK1, ATG9, LC3, VMP1, and GABARAP, increasing their transcription levels [78]. In contrast, SIRT5-mediated deacetylation of LDHB was found to induce autophagy [79]. Increased acetylation by Acetyl-CoA represses the expression of the autophagy gene, ATG7, and a high level of methylation by the methyltransferase, G9A, can inhibit the transcription levels of ATG9, LC3, GABARAP, BNIP3, SQSTM1, and WIPI1 [77]. Epigenetic and transcriptional regulation of autophagy genes by histone modification impairs the formations of autophagosomes and autolysosomes and reduces the LC3 flux in cellular senescence and aging [64].

## 4. QC of RNAs and Proteins in Age-Related Diseases

Accumulating evidence indicates that the decline in RNA and protein QC contributes to many age-related diseases, such as cancer, neurodegenerative diseases, muscle diseases, and cardiovascular diseases. In this part of the review, we summarize what is known about alterations of RNA and protein QC in cellular senescence and aging.

### 4.1. Cancer

NMD, which is a highly conserved RNA decay pathway, plays dual roles in cancer. First, it can act as a tumor-promoting pathway by degrading suppressor genes. In a tumor, the tumor suppressor genes have a higher ratio or insertion and deletion (indel) and nonsense mutations relative to missense mutations [80]. Analysis of genome-wide expression data reveals that tumor suppressor genes, such as EIF5B, LARP4B, and PTEN, p53, have PTC-inducing mutations, which are also called NMD-eliciting mutations [81,82,83,84]. UPF1, a core component of NMD, also degrades the *p21* mRNA by cooperating with other factors, such as the *Linc-ASEN* lncRNA and the mRNA repressor, TRIM71 [85,86]. In cancer cells, *Linc-ASEN* lncRNA interacts with UPF1 and DCP1A and induces *p21* mRNA decay. Depletion of *Linc-ASEN* lncRNA or UPF1 increases the *p21* mRNA level to mediate cellular senescence in cancer cells. *Linc-ASEN* has inverse correlation p21 expression in tumors from patient tissues and patient-derived mouse xenograft. In addition, expression of *Linc-ASEN* is significantly lower in tissues from aged mouse than young mouse, and its expression shows negative correlation with *p21* mRNA in mouse tissues [85]. TRIM71 is upregulated in various cancer types, such as hepatocellular carcinoma, acute myeloid leukemia and ovarian cancer. In hepatocellular carcinoma cells, TRIM71 reduces *p21* mRNA levels and promotes cancer cell proliferation. TRIM71 directly interacts with the stem loop in the *p21* 3′ UTR and induces NMD-mediated *p21* mRNA decay by cooperating with NMD factors, UPF1 and SMG1 [86]. Wig1 (also referred to as ZMAT3) induces *p21* mRNA decay via the recruitment of Ago2, a major component of the RNA-induced silencing complex (RISC). Depletion of Wig1 stabilizes the *p21* mRNA, resulting in the induction of growth arrest and cellular senescence in breast, colon and lung cancer cells. Moreover, in mouse xenograft model, Wig1 depletion repress tumor growth by increasing p21 levels [87]. In cancer, oncogenes such as the pro-survival gene, BCL2, can escape NMD. BCL2 has an immunoglobulin-specific sequence (NMD-escape mutation) in its 3′ UTR; this prevents its NMD-mediated mRNA degradation, resulting in a high level of BCL2 in cancer [88]. NMD can also suppress tumor progression via the degradation of mutated tumor suppressor mRNAs (*p53*), oncogene mRNAs, noncoding RNAs (*MALAT*, *SNHG5*), and an EMT-associated mRNA (*TWIST*). The mechanisms by which NMD suppresses tumors still remain to be studied. HuD (also called ELAVL4), which is an RNA-binding protein (RBP) belonging to the human antigen Hu/ELAVL family, regulates gene expression at the post-transcriptional level by modulating alternative splicing, mRNA stability, mRNA localization, and translation. In human neuroblastoma cells, HuD negatively regulates the expression of senescence-associated secreted proteins (SASPs), including IL-6, CXCL2, CCL20, and CCL2. Depletion of HuD increases the mRNA levels of SASPs. HuD has been shown to directly bind to the 3′ UTR of the *CCL2* mRNA, but the underlying mechanism remains to be elucidated in detail [89].

The UPS plays crucial roles in regulating protein QC and homeostasis. Impairment of UPS function is involved in cancer; in numerous cancers, dysregulation of ubiquitination is caused by epigenetic alteration (hypermethylation), genetic mutation in coding regions, post-transcriptional modification (aberrant splicing or destabilization of mRNA by miRNAs) or post-translational modification (phosphorylation or self-ubiquitination). E3 ligase overexpression is also implicated in the growth and survival of cancer [90]. TRIM25 is highly expressed in HCC tissues relative to adjacent normal tissues. In colon cancer cells, TRIM25 expression is increased upon ER stress and upregulated TRIM25 reduces Keap1, an inhibitor of Nrf2, by its ubiquitination and degradation. Decreased Keap1 by TRIM25 activates Nrf2 and leads to tumor cell proliferation. In HCC xenograft mouse models, depletion of TRIM25 increases Keap1 and inactivates Nrf2, suppressing tumor growth [91]. TRIM32 excessively degrades ARID1A and induces proliferation of human esophageal squamous carcinoma cells. Reduced ARID1A by TRIM32 increases tumor growth and chemoresistance in squamous cell carcinoma xenograft mouse model [92]. Skp2 is a component of the Skp2-Culin-2-F-box (SCF) E3 ligase complex. Skp2 is overexpressed in many types of human cancers, such as breast cancer, non-small cell lung cancer (NSCLC), lymphoma, melanoma, pancreatic cancer, and prostate cancer. Its overexpression is correlated with poor survival, increased resistance to anti-cancer drugs, and adverse clinical outcomes of patients [93]. Skp2 regulates the proliferation, apoptosis, migration, invasion, angiogenesis, and metastasis of cancer cells [94,95,96,97,98,99,100,101]. It promotes the degradation of p21 and p27 through K48-specific ubiquitination to increase cell proliferation, migration, and invasion [94,95,96,97,98,99]. Skp2 inhibits apoptosis by inducing the polyubiquitination and degradation of FOXO1 in prostate cancer [100] and induces the phosphorylation and ubiquitin-mediated degradation of PDCD4 in breast cancer to suppress apoptosis and increase cell proliferation and radiotherapy resistance [101]. MDM2 is a negative regulator of p53 and its overexpression is frequently observed in cancers. MDM2 mediates the proteasomal degradation of p53, leading to therapeutic resistance of tumors [102]. The E3 ligase NEDD4-1 plays dual roles as an oncogene and tumor suppressor in cancers. Degradation of PTEN and CNrasGEF or stabilization of MDM2 by NEDD4-1 promotes cell proliferation, migration, and invasion. In contrast, degradation of N-Myc, Her3, and RAS by NEDD4-1 inhibits cell proliferation and tumorigenesis. NEDD4-1 overexpression induces apoptosis through the degradation of SAG in cancer cells [103]. FBXW7 is a member of the SCF E3 ligase complex and functions as a tumor suppressor. FBXW7 is inactivated by genomic deletion, genetic mutation and hypermethylated promoter in human colorectal, gastric, and breast cancers [104]. Its loss is associated with increased invasion and metastasis of cancer and poor survival of patients [105,106,107]. FBXW7 induces the ubiquitin-mediated degradation of oncoproteins, including cyclin E, Aurora A, Notch1, mTOR, c-Myc, Mcl-1, and Jun [108].

Autophagy generally inhibits the neoplastic transformation from normal healthy cells. Conversely, however, autophagy supports neoplastic transformation, cancer growth, tumor relapse, and therapeutic resistance depending on the cancer type and stage. In its tumor suppression function, autophagy represses tumor initiation, growth, and development by increasing antitumor immunity. However, in several cancers, autophagy promotes tumor initiation, progression, metastasis, and therapeutic resistance [109]. TRIM59 E3 ligase is overexpressed in metastatic breast cancer and its upregulation is closely associated with cancer cell survival and metastasis. TRIM59 increases the stability of PDCP10 by preventing RNFT1-mediated K63 ubiquitination and SQSTM1-mediated autophagic degradation. Stabilized PDCD10 inhibits RHOA-ROCK downstream signaling and actomyosin-mediated contractility, leading to tumor formation and metastasis [110].

### 4.2. Neurodegenerative Diseases

#### 4.2.1. Alzheimer’s Disease (AD)

Alzheimer’s disease (AD) is the most prevalent form of dementia and is closely correlated with cellular senescence and aging. It is characterized by the accumulation of amyloid-β (Aβ) plaques and phosphorylated tau tangles, which causes memory loss and amnestic cognitive impairment [111].

Aβ accumulation can be regulated by RNA QC of amyloid precursor protein (*APP*) and β-site AβPP cleaving enzyme 1 (*BACE1*) [112,113,114,115,116,117,118]. β-secretase 1 encoded by *BACE1* cleaves APP and induces the formation of abnormal Aβ proteins and aggregates [119]. The *APP* and *BACE1* mRNAs are increased in the aged cerebral brain [112,113]. They are regulated by the RBP, HuD, which is increased in the brain tissues of AD patients. HuD stabilize the *APP* mRNA, *BACE1* mRNA, and *BACE1-AS* lncRNA in neuroblastoma cells, leading to Aβ accumulation [114]. *BACE1* mRNA and its encoded protein levels are regulated by *BACE1-AS*, which increases *BACE1* mRNA stability by preventing its miRNA-mediated degradation [115]. QC of the *APP* mRNA is regulated by FMRP and hnRNPC via mRNA decay. FMRP, a fragile X mental retardation protein, recruits the *APP* mRNA into processing-bodies (P-bodies) to inhibit its translation. Conversely, the increased binding of hnRNPC enhances APP translation. In neuroblastoma cells, the FMRP level is reduced and hnRNPC can bind to the *APP* mRNA, resulting in an increase of the APP protein level [116,117,118].

Impaired protein QC by the UPS and autophagy is also linked to AD. The UPS plays an important role in preventing the accumulations of Aβ and tau. RNF182, a brain-enriched E3 ligase, is upregulated in tissues from AD brains. RNF182 is increased in neurons and astrocytes treated with oxygen and glucose deprivation (OGD). Upregulation of RNF182 promotes the degradation of ATP6V0C, which is a key integral protein of gap junctions and neurotransmitter release, resulting in impaired function of gap junction and increased neuronal cell death [120]. The deubiquitinase, USP11, accumulates in brain tissues from AD and frontotemporal lobar degeneration with τ pathology (FTLD-tau). In HeLa cells stably expressing wildtype tau, UPS11 overexpression increases stability and aggregation of tau via its DUB catalytic activity. USP11 is involved in the process of tau acetylation (K281 and K274) in cultured cells. In addition, the expression and accumulation levels of USP11 are elevated in females’ mice and people than in males, exhibiting the high level of tauopathy signatures and tau tangle density [121]. NRBP1, a substrate receptor of Cullin-RING ubiquitin ligase (CRL), promotes the degradation of BRI2 and BRI3, which are negative regulators of Aβ production. In neuronal cells, depletion of NRBP1 reduces Aβ production by increasing the abundance of BRI2 and BRI3 [122]. FKBP51, which is a co-chaperone with Hsp90, is highly expressed in the aged and AD brain. FKBP51 gene is demethylated and its expression is increased with aging and AD progression. In mouse model, FKBP51 and Hsp90 blocks the clearance of tau and induces tau oligomerization, increasing tau toxicity [123]. The E3 ligase, CHIP, is enriched in brain and its downregulation is associated with the accumulations of Aβ and tau in AD patients [124,125]. In neuronal cells, CHIP increases the degradation of Aβ42 peptide and phosphorylated tau to decrease neuronal toxicity [126,127]. CHIP promotes the ubiquitin-mediated degradation of β-secretase 1 to reduce APP processing and Aβ levels. Moreover, the p53-mediated stabilization of CHIP represses BACE1 transcription and APP processing in neurons and HEK-APP cells [124].

PINK1 functions to degrade dysfunctional mitochondria via mitophagy. It is decreased in the brain of patients with AD and in a transgenic AD mouse model. Rescue of PINK1 in neuronal cells decreases Aβ accumulation and its associated pathology [128]. Presenilin-1 is a catalytic subunit of the γ-secretase complex, which cleaves βCTF to produce Aβ. Mutations of Presenilin-1, which are found in the familiar AD, increase the ratio of Aβ42 to Aβ40. Increased generation of Aβ42 relative to Aβ40 is a major component of amyloid plaques in the AD brains [129]. The autophagy-related protein, Beclin-1, is decreased in the brain of AD patients. Beclin-1 mRNA and protein levels are also reduced in human aged brain, independent of AD pathology. Decreased Beclin-1 expression impairs autophagosome formation and leads to the accumulation of Aβ [130]. CCT (also known as TRiC) is a member of the chaperonin family; it is required for autophagy processes, autophagosome formation, and lysosomal fusion, and thereby prevents the aggregation of neuropathogenic proteins [131]. CCT expression is reduced in aging and neurodegenerative diseases [132]. Decreased CCT promotes LC3 degradation, improper autophagosome formation, and tau folding [131,132,133]. Finally, increased tau tangles repress the autophagic flux and disrupt autophagosome maturation via HDAC6 inhibition [134,135].

#### 4.2.2. Parkinson’s Disease (PD)

Parkinson’s disease (PD) is an age-related neurodegenerative disease caused by loss of dopaminergic neurons. This degeneration is correlated with the increased aggregation of α-synuclein and the formation of Lewy bodies [136].

ATP13A2 (also known as PARK9) is a key gene in PD; its mutation was first found in a recessive form of PD. ATP13A2 mutations can be divided into nonsense or missense mutations. The product of the nonsense-mutated ATG13A2 gene has PTC and activates the NMD pathway. The nonsense *ATP13A2* mRNA is degraded and the corresponding protein level is reduced in PD. Missense-mutated ATG13A produce the misfolded ATP13A2 protein. Misfolded proteins are mislocalized to endoplasmic reticulum (ER) and induces endoplasmic reticulum-associated degradation (ERAD) pathway, leading to a decrease protein level of ATP13A2. Loss-of-function mutation in ATP13A2 leads to increased aggregation and limited secretion and extracellular clearance of α-synuclein. Moreover, a decreased ATP13A2 protein level reduces the uptake of α-synuclein by astrocytes and increased the accumulation of α-synuclein in the dopaminergic cells, resulting in its pathological accumulation [137]. The accumulated α-synuclein protein can regulate P-bodies and mRNA stability. α-Synuclein closely interacts with core proteins of mRNA decapping and degradation, such as Edc3, Edc4, Dcp1, Dcp2, and Xrn1, and negatively regulates mRNA stability in PD [138].

Dysfunctions of the UPS and autophagy caused by mutations of associated proteins increase the levels of α-synuclein and PD-related aggregates. UCHL1 (also known as PARK5) is a DUB enzyme and is highly expressed in neurons. In German familial PD, UCHL1 has missense mutation, an I93M substitution [139]. In cells and mouse models, UCHL1^I93M^ mutant has highly affinity for LAMP-2A and blocks CMA-mediated degradation of α-synuclein, increasing nigral cell death [140,141]. UCHL1 is a target of oxidative damage and is modified by oxidation in aging. Oxidized UCHL1 interact with LAMP-2A, Hsp90 and Hsc70 and blocks CMA dependent degradation of α-synuclein in cells [140,142]. Mutations of LRRK2 are closely correlated with PD. Mutations (I1371V, R1441G, Y1699C, G2019S, I2012T and I2020T) in Rasp of complex proteins (ROC domain), C-terminal of Roc (COR domain) and kinase domains of LRRK2 activates their kinase activities. These mutations of LRRK2 cause the abnormal mitochondrial dynamics and the aberrant autophagic-lysosomal pathway, leading to cell death and α-synuclein accumulation [143]. Nitrosative stress is increased in aging and PD. Nitrosative stress induces S-nitrosylation of PINK1 and Parkin, and these protein modifications disrupt mitophagy, accumulating damaged mitochondria. In addition, loss of function mutations in PINK1 and Parkin genes are major cause of early-onset PD [144].

#### 4.2.3. Huntington’s Disease (HD)

Huntington’s disease (HD) is a neurodegenerative disease caused by expansion of the CAG trinucleotide repeat in the huntingtin gene (HTT). Mutant huntingtin (mHTT) has an abnormally long polyglutamine tract and exhibits protein toxicity, leading to neuronal dysfunction and death. In CMA (a type of autophagy), a chaperone protein selectively recognizes KFERQ-like motif-bearing mHTT and forms a complex between Hsc70, mHTT, and co-chaperones. This complex is delivered to the lysosomal membrane and interacts with the CMA receptor, LAMP2A. mHTT is completely degraded in the lumen of the lysosome. However, when cargo recognition is defective, mHTT is accumulated in HD [145,146].

### 4.3. Muscle Diseases

Poly(A) binding protein nuclear 1 (PABPN1) is a multifunction regulator of RNA stability. PABPN1 is decreased in muscles from oculopharyngeal muscular dystrophy (OPMD) patients and aged people, and its reduced level is associated with muscle weakness [147]. PABPN1 regulates both distal polyadenylation site (PAS) utilization and proteasomal activity. Decreased PABPN1 levels are associated with reduced distal PAS utilization in the 3′ UTR of the OPMD-associated gene, Atrogin-1, resulting in its upregulation. Downregulated PABPN1 also decreases proteasomal degradation and contributes to altering the MyHC isotype pattern, leading to muscle fiber-type transition [148].

Skeletal muscle satellite cells (MuSCs) play a crucial role in muscle repair and maintenance by undergoing division and differentiation. The regenerative potential of MuSCs decreases with age, and their dysfunction is associated with the age-related muscle loss known as sarcopenia [149,150]. The p38 α/β MAPK pathway induces the transcriptional activities of MyoD and MEF2, leading to the myogenic differentiation of MuSCs. Tristetraprolin (TTP), an mRNA decay factor, destabilizes the 3′ UTR of the *MyoD* mRNA, which encodes a myogenic transcription factor and thereby prevents the activation of MuSCs [151]. TTP binds to AU-rich elements within the 3′ UTR of target transcripts and recruits mRNA decay enzymes such as the deadenylase components, Not1 and Caf1 [152]. AU-rich mRNA binding factor 1 (AUF1), an mRNA decay factor, is decreased in the skeletal muscle of older mice. Gene transfer of muscle-specific AUF1 using an adeno-associated virus (AAV) system increases muscle mass, exercise endurance, and satellite cell activation, and reduces markers of muscle atrophy. AUF1 gene transfer increases the expression of pgc1a, which is a regulator of mitochondrial biogenesis and oxidative metabolism, by stabilizing the *pgc1a* mRNA. These findings indicate that AUF1 regulates the maintenance and differentiation of MuSCs through the regulation of mRNA stability [153].

Skeletal muscle atrophy is caused by various conditions, including oxidative stress, inflammation, and malnutrition. These stimuli activate transcription factors, including FOXO1/FOXO3a, NF-kB, KLF15, Smad 3, and glucocorticoid receptor (GR), to increase the transcription levels of muscle-specific E3 ubiquitin ligases, such as MuRF1 and Atrogin-1. MuRF1 and Atrogin-1 are crucial for the regulation of protein degradation in skeletal muscle, and are upregulated in skeletal muscle atrophy [154]. MuRF1 may regulate MHC-related proteins to induce overall protein degradation, whereas Atrogin-1 degrades eIF3-f, myogenin, and MyoD to affect protein synthesis and myogenic differentiation [155]. The expression levels of MuRF1 and Atrogin-1 are significantly increased in the muscles of aged rats [156]. Another E3 ubiquitin ligase, tumor necrosis factor (a) receptor adaptor protein 6 (TRAF6), has been found to be associated with skeletal muscle atrophy. TRAF6 is upregulated in skeletal muscle atrophy, and its inhibition suppresses myosin heavy chain proteolysis and rescues muscle atrophy in denervated skeletal muscles [157]. In mice model, targeted ablation of TRAF6 prevents the expression of muscle atrophy regulators, such as Beclin1, p62, and LC3B, in starvation-induced muscle atrophy, and skeletal muscle-specific TRAF6 knockout rescues the muscle atrophy induced by denervation and starvation [157,158].

Autophagy in skeletal muscle is tightly regulated by diverse pathways, including AMPK/mTOR, GR, and NF-kB signaling. It is crucial for generating and consuming energy in skeletal muscle, and autophagy dysfunction can lead to skeletal muscle disease by inducing cellular alterations, such as cell death, mitochondrial damage, and ER stress. Autophagy must be properly controlled to maintain cellular homeostasis in skeletal muscle [159]. In muscle diseases, impaired mitophagy (i.e., the removal of dysfunctional mitochondria) induces muscle loss and weakness. Several autophagy/mitophagy-related genes, including Beclin-1, BNIP3, Parkin, and ATG7, are downregulated in skeletal muscle of older women. It has been suggested that autophagy/mitophagy-related genes are correlated with decreased physical function in frail elderly women, reflecting impairment of mitochondrial removal [160]. Atg7-deficient mice show muscle atrophy, inflammation, and muscle weakness due to enhancements of mitochondrial dysfunction and oxidative stress [161].

### 4.4. Cardiovascular Diseases

Cardiac aging is associated with cardiovascular diseases, including fibrosis, hypertension, stroke, and atherosclerosis. Atherosclerosis is an age-related cardiovascular disease caused by cholesterol plaques. The dysfunction of endothelial cells (ECs) leads to high-level expression of adhesion molecules such as VCAM-1 for the progression of atherosclerotic plaques [162]. *miR-21* can affect the dysfunction of ECs and the formation of atherosclerosis. *miR-21* induced by oscillatory shear stress represses *PPARa* by directly binding to its 3′ UTR, and thereby increases the expression of VCAM-1 and MCP-1. These factors promote the dysfunction of the ECs and the initiation of atherosclerotic plaques [163]. Conversely, downregulation of *miR-126* increases proinflammatory *TNF-α*, which stimulates the activity of NF-kB and VCAM-1. These effects induce EC dysfunction and promote atherosclerosis [162]. The RBP, zinc finger protein 36, is increased in coronary arteries from patients with atherosclerosis. Upregulation of ZFP36 decreases inflammation in vascular endothelial cells by negatively regulating the mRNA stability of cytokines, including *MCP-1* and *IL-6*. ZFP36 also represses the transcriptional activation of NF-kB by reducing its nuclear import, suggesting that ZFP36 is a potential target for preventing atherosclerosis [164].

The E3 ubiquitin ligase, NEDD4L, is involved in the development of both essential and salt-sensitive hypertension [165]. In salt-sensitive hypertension, glucocorticoid regulated kinase 1 (SGK1) phosphorylates and inactivates NEDD4L, finally disrupting NEDD4L-mediated degradation of epithelial sodium channel (ENaC). The abnormal regulation of ENaC contributes to the development of salt-sensitive hypertension [166]. SMAD-specific E3 ubiquitin protein ligase 1 (SMURF1) is associated with pulmonary arterial hypertension (PAH). The expression level of SMURF1 is upregulated in blood samples of patients with PAH. Reduced BMP signaling is a pathological finding in PAH, and the downregulation of SMURF1 stimulates BMP signaling by inhibiting BMPR2 degradation in vitro and in preclinical models [167].

Autophagy and mitophagy are important for maintaining cardiovascular homeostasis. Autophagy decreases with age, and mitochondrial dysfunction is increased in aging heart. Therefore, activation of autophagy inhibits cardiovascular aging and increases the lifespan [168]. Although the mechanisms of age-related decreases in autophagy remain unknown, it may be reasonable to address it by manipulating the longevity pathways, including mTOR, AMPK, IGF-1/PIK2/Akt, and Sirtuin signaling. Several studies involving the depletion of autophagy-related genes in animal models show the importance of autophagy and mitophagy in cardiovascular homeostasis and diseases [169]. In general, suppression of autophagy promotes cardiovascular aging and diseases: Atg5-deficient mice exhibit cardiovascular diseases, including cardiac hypertrophy, contractile dysfunction, and left ventricular dilatation [170]. Knockout of Pink1, a mediator of mitophagy, induces cardiac hypertrophy and left ventricular dysfunction through impaired mitochondrial biogenesis in mice [171]. Mice with cardiomyocyte-specific depletion of the mitophagy-relevant protein, Mnf2, show respiratory dysfunction and cardiomyopathy due to inhibition of mitochondrial QC [172]. Conversely, activation of autophagy by genetic, nutrient, or pharmacological stimulations might decelerate cardiovascular aging and disorders. Cardiomyocyte-specific overexpression of Parkin improves cardiac function and reduces the positivity of senescence-associated β-galactosidase (SA-β-Gal) through increasing Parkin-mediated mitophagy in aged mice [173]. Overexpression of Atg5 activates autophagy, delays cardiac aging, and extends the lifespan in mice [174]. Treatment of rapamycin promotes autophagy and inhibits cardiac hypertrophy through inhibition of mTOR [175].

## 5. Conclusions

QC (QC) of RNAs and proteins is key biological process in maintaining physiological and cellular homeostasis. Increasing evidence supports the idea that QC of RNAs and proteins is dysregulated during cellular senescence and contributes to multiple age-related diseases, such as cancers and neurodegenerative, metabolic, muscle, and cardiovascular diseases. The results of the existing studies reveal that diverse mechanisms of RNA and protein QC are associated with aging (Table 1). However, open questions remain regarding the molecular targets, mechanisms, and signaling pathways of RNA and protein QC in cellular senescence and aging. In this review, we summarize the molecular targets of RNA and protein QC in cellular senescence, aging, and age-related diseases, such as cancer and neurodegenerative, muscle, cardiovascular diseases. Continued discovery of novel targets and further understanding of the molecular mechanisms of RNA and protein QC in cellular senescence and aging will be necessary for the development of effective therapies against age-related diseases.

## Figures and Tables

**Figure 1 cells-11-04062-f001:**
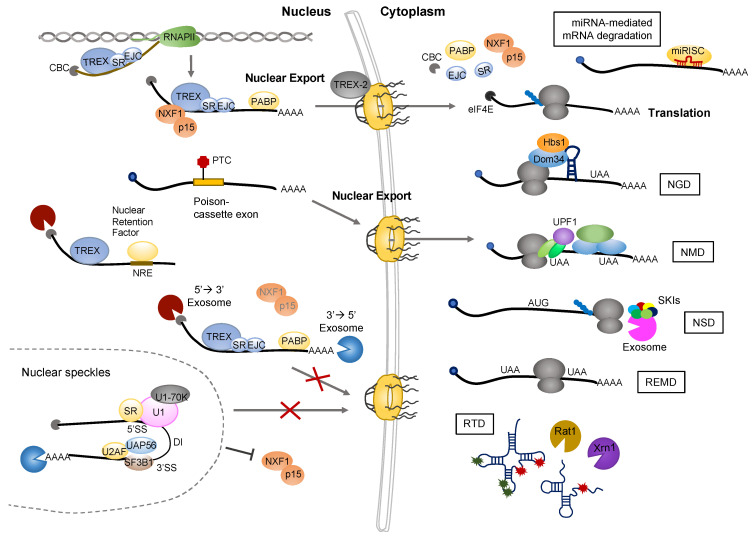
Molecular mechanisms of RNA QC. In eukaryotic cells, RNA biogenesis has multiple series of processes, involving transcription, 5′ capping, splicing, polyadenylation, and export to cytoplasm, and translation. During these processes, defective RNAs are subjected to RNA QC in the nucleus and cytoplasm. In nucleus, mRNAs with ‘detained’ introns are degraded by nuclear exosomes in the nuclear basket. mRNAs, which interacts with retention factors or lacks export factors, are retained in the nucleus and finally degraded by nuclear exosomes. ‘Poison-cassette” exon-contained mRNAs, which have PTC in their exon, are exported to the cytoplasm and degraded by NMD pathway. In cytoplasm, abnormal RNAs can be distinguished from normal RNAs by RNA structure and binding proteins, and are directly degraded by several degradation pathways. mRNAs with PTC or long 3′UTR are degraded by NMD. In a process of NGD, mRNAs with translation elongation stalled structure are degraded by recruiting Dom34–Hbs1 complex. NSD degrades mRNAs having no stop codon, and REMD destabilized mRNA in a poly(A)-dependent manner. In a process of RTD, tRNAs lacking several body modifications are degraded by 5′-3′ exonuclease, Rat1 and Xrn1.

**Figure 2 cells-11-04062-f002:**
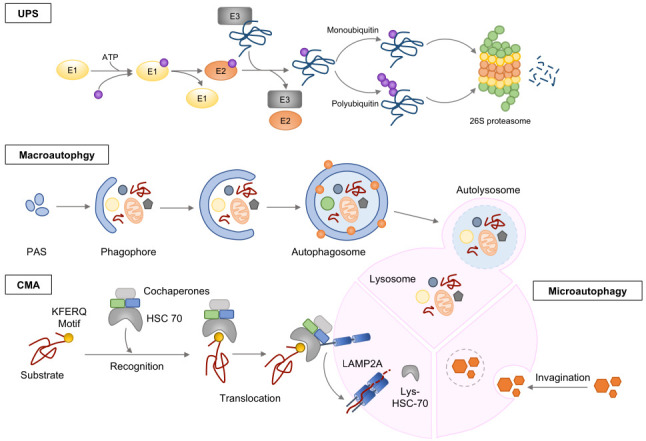
Various mechanisms of protein QC. Two representative protein QC systems, UPS and autophagy, maintain the protein homeostasis. Misfolded or abnormal proteins are degraded through UPS. Mono or polyubiquitinated proteins by the ubiquitin-activating enzymes, E1, E2 and E3, are recruited to 26S proteasome for degradation. The other protein QC system, autophagy, has three major forms of autophagy, macroautophagy, microautophagy, and chaperon-mediated autophagy (CMA). Macroautophagy sequesters cytosolic cargos within double-membrane vesicles called autophagosomes, and subsequently transfers them to lysosomes for degradation. Microautphagy directly uptake cargos through membrane protrusion and either invagination of the lysosomal or late endosomal membranes. CMA transports cargos with KFERQ-like motif to the lysosome membranes. These three types of autophagy degrade cargos and maintain proper protein homeostasis in the cells.

**Table 1 cells-11-04062-t001:** Molecular pathways of RNA and proteins related to age-related diseases.

RNA or Proteins	Mechanisms of Regulation	References
**Cancer**		
*Linc-ASEN*	Degradation of *p21* mRNA by cooperating with NMD factors, UPF1 and DCP1A.	[85]
TRIM71	NMD-mediated *p21* mRNA decay with NMD factors, UPF1 and SMG1	[86]
Wig1	Destabilize *p21* mRNA by recruiting Ago2, a major component of RISC	[87]
HuD/ELAVL4	Negatively regulation of *CCL2* mRNA by directly binding to 3’UTR of the *CCL2* mRNA	[89]
TRIM25	Reduction of Keap1, a Nrf2 inhibitor, by its ubiquitination and degradation	[91]
TRIM32	Decrease ARID1A protein level via ubiquitin-mediated degradation	[92]
Skp2	Degradation of p21, p27, FOXO1 and PDCD4 through ubiquitination	[93,94,95,96,97,98,99,100,101]
MDM2	Ubiquitination and degradation of p53	[102]
NEDD4-1	Degradation of PTEN, CNrasGEF, N-Myc, Her3 and Ras via K48-linked ubiquitination. Stabilization of MDM2 via K63-linked ubiquitination	[103]
FBXW7	Ubiquitin-mediated degradation of cyclin E, Aurora A, Notch1, mTOR, c-Myc, Mcl-1 and Jun	[104,105,106,107,108]
TRIM59	Increase the stability of PDCD10 by inhibiting RNFT1-mediated K63 ubiquitination and SQSTM1-mediated autophagic degradation	[110]
**Neurodegenerative diseases-Alzheimer’s diseases (AD)**		
HuD/ELAVL4	Increase the stability of *APP* mRNA, *BACE1* mRNA, and *BACE1-AS* lncRNA	[114]
FMRP	Recruiting the *APP* mRNA into P-bodies to inhibits its translation.	[116,117,118]
hnRNPC	Binding to *APP* mRNA and enhancing APP translation.	[116]
RNF182	Ubiquitination and degradation of ATP6V0C	[120]
USP11	Increase of tau stability and aggregation. Involve in the process of tau acetylation (K281 and K274)	[121]
NBRP1	Degradation of BRI2 and BRI3	[122]
FKBP51	Inhibition of tau clearance and increase of tau aggregation by cooperating with Hsp90.	[123]
CHIP	Ubiquitin-mediated degradation of Aβ42 peptide, phosphorylated tau and β-secretase 1	[124,125,126,127]
PINK1	Degradation of dysfunctional mitochondria via mitophagy	[128]
Presenilin-1	Cleaves of βCTF to produce Aβ.	[129]
Beclin-1	*Beclin-1* mRNA and protein levels are also reduced in human aged brain, independent of AD pathology. Decreased Beclin-1 expression impairs autophagosome formation and leads to the accumulation of Aβ	[130]
CCT/TRiC	CCT expression is reduced in aging and neurodegenerative diseases. Decreased CCT promotes LC3 degradation, improper autophagosome formation, and tau folding	[131,132,133]
**Neurodegenerative diseases-Parkinson’s disease (PD**)		
ATP13A2/PARK9	Preventing the accumulation and aggregation of α-synuclein in the dopaminergic cells	[137]
UCHL1/PARK5	Role as deubiquitinase or hydrolase enzymes in the UPS.	[139,140,141,142]
LRRK2	Mutation of LRRK2 cause the abnormal mitochondrial dynamics and the aberrant au-tophagic-lysosomal pathway, leading to cell death and α-synuclein accumulation	[143]
PINK1, Parkin	S-nitrosylation of PINK1 and Parkin disrupt mitophagy, accumulating damaged mitochondria.	[144]
**Neurodegenerative diseases-Huntington’s disease (HD)**		
HTT	Modulator of selective autophagy. Required for autophagy recognition and activating machinery (p62 and ULK1).	[145,146]
**Muscle disease**		
PABPN1	Regulation of distal polyadenylation site and proteasomal activity	[147,148]
TTP	Destabilize *MyoD* mRNA by recruiting the deadenylase components, Not1 and Caf1	[151,152]
AUF1	Maintenance and differentiation of MuSCs via the regulation of mRNA stability	[153]
MuRF1	Regulation of MHC-related proteins to induce overall protein degradation	[154,155,156]
Atrogin-1	Degradation of eIF3-f, myogenin, and MyoD to affect protein synthesis and myogenic differentiation	[154,155,156]
TRAF6	Prevention of Beclin1, p62, and LC3B through ubiquitination	[157,158]
ATG7	Atg7-deficient mice induces muscle atrophy, inflammation, and muscle weakness by mitochondrial dysfunction and oxidative stress	[160,161]
**Cardiovascular disease**		
*miR-21*	Repression of *PPARa* mRNA by directly binding to 3’UTR	[162,163]
*miR-126*	Repression of proinflammatory *TNF-α* mRNA	[162]
ZFP36	Regulation of *MCP-1* and *IL-6* mRNA stability	[164]
NEDD4L	Degradation of epithelial sodium channel (ENaC) through ubiquitination	[165,166]
SMURF1	Degradation of BMPR2 through ubiquitination	[167]
ATG5	Atg5-deficient mice exhibit cardiovascular diseases, including cardiac hypertrophy, contractile dysfunction, and left ventricular dilatation. Overexpression of Atg5 activates autophagy, delays cardiac aging, and extends the lifespan in mice	[170,174]
PINK1	Knockout of Pink1 induces cardiac hypertrophy and left ventricular dysfunction through impaired mitochondrial biogenesis in mice	[171]
Parkin	Cardiomyocyte-specific overexpression of Parkin improves cardiac function and reduces the positivity of senescence-associated β-galactosidase (SA-β-Gal) through increasing mitophagy in aged mice	[173]
Mnf2	Protein required for mitochondrial dynamics and associated with mitophagy	[172]

## Data Availability

Not applicable.

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
