# Peer review of "Mechanisms of RNA and Protein Quality Control and Their Roles in Cellular Senescence and Age-Related Diseases"

_cells, 2022, doi:10.3390/cells11244062_

Round 1
Reviewer 1 Report
The paper is interesting, accurate and well written. The figures help the reader. I have only minor suggestions, such as to check the style, since liness 199-219 are different from the rest of the paper. Morevover, I suggest the addition of some tables to summarize the molecules/pathways more frequently involved in the different diseases.
The review by Kang and coll. is really detailed in describing the role of RNA quality control in different fields, specifically in senescence and neurodegenerative diseases. It offers an accurate description of the different mechanisms of RNA quality control in the cells, and then it relates them to senescence and age-related diseases. In my opinion, a point of strength is that it covers the main aspects of the issue with an adequate level of description and with also some schematic figures. I really don’t have a request for new paragraphs, I just suggest i) to add a table to recap the molecular pathways involved in neurodegenerative diseases, in order to identify the points of similarity, ii) to adjust the style. I can add as a minor observation that for the expression “Quality control” in some cases the authors used the acronym QC while in others they don’t, without a logic, but it is the only thing, for the rest, it is a well written and clear review.
Author Response
Response to Reviewer 1 Comments
Point 1: The paper is interesting, accurate and well written. The figures help the reader. I have only minor suggestions, such as to check the style, since liness 199-219 are different from the rest of the paper. Morevover, I suggest the addition of some tables to summarize the molecules/pathways more frequently involved in the different diseases.
Response 1: We appreciate your comment. As suggested, we checked and revised the style of lines 199-219. We also added a table to summarize the molecules/pathways in the different diseases (Table 1).
Point 2: The review by Kang and coll. is really detailed in describing the role of RNA quality control in different fields, specifically in senescence and neurodegenerative diseases. It offers an accurate description of the different mechanisms of RNA quality control in the cells, and then it relates them to senescence and age-related diseases. In my opinion, a point of strength is that it covers the main aspects of the issue with an adequate level of description and with also some schematic figures. I really don’t have a request for new paragraphs, I just suggest i) to add a table to recap the molecular pathways involved in neurodegenerative diseases, in order to identify the points of similarity, ii) to adjust the style. I can add as a minor observation that for the expression “Quality control” in some cases the authors used the acronym QC while in others they don’t, without a logic, but it is the only thing, for the rest, it is a well written and clear review.
Point 2-1: I just suggest i) to add a table to recap the molecular pathways involved in neurodegenerative disease
Response 2-1: We appreciate your valuable comment. As suggested, we added a table (Table 1) to recap the molecular pathways involved in age-related diseases, including cancer, neurodegenerative diseases, muscle disease, and cardiovascular disease.
Table 1. Molecular pathways of RNA and proteins related to age-related diseases
|
RNA or proteins |
Mechanisms of regulation |
References |
|
Cancer |
||
|
Linc-ASEN |
Degradation of p21 mRNA by cooperating with NMD factors, UPF1 and DCP1A. |
[85] |
|
TRIM71 |
NMD-mediated p21 mRNA decay with NMD factors, UPF1 and SMG1 |
[86] |
|
Wig1 |
Destabilize p21 mRNA by recruiting Ago2, a major component of RISC |
[87] |
|
HuD/ELAVL4 |
Negatively regulation of CCL2 mRNA by directly binding to 3'UTR of the CCL2 mRNA |
[89] |
|
TRIM25 |
Reduction of Keap1, a Nrf2 inhibitor, by its ubiquitination and degradation |
[91] |
|
TRIM32 |
Decrease ARID1A protein level via ubiquitin-mediated degradation |
[92] |
|
Skp2 |
Degradation of p21, p27, FOXO1 and PDCD4 through ubiquitination |
[93-101] |
|
MDM2 |
Ubiquitination and degradation of p53 |
[102] |
|
NEDD4-1 |
Degradation of PTEN, CNrasGEF, N-Myc, Her3 and Ras via K48-linked ubiquitination. Stabilization of MDM2 via K63-linked ubiquitination |
[103] |
|
FBXW7 |
Ubiquitin-mediated degradation of cyclin E, Aurora A, Notch1, mTOR, c-Myc, Mcl-1 and Jun |
[104-108] |
|
TRIM59 |
Increase the stability of PDCD10 by inhibiting RNFT1-mediated K63 ubiquitination and SQSTM1-mediated autophagic degradation |
[110] |
|
Neurodegenerative diseases-Alzheimer's diseases (AD) |
||
|
HuD/ELAVL4 |
Increase the stability of APP mRNA, BACE1 mRNA, and BACE1-AS lncRNA |
[114] |
|
FMRP |
Recruiting the APP mRNA into P-bodies to inhibits its translation. |
[116-118] |
|
hnRNPC |
Binding to APP mRNA and enhancing APP translation. |
[116] |
|
RNF182 |
Ubiquitination and degradation of ATP6V0C |
[120] |
|
USP11 |
Increase of tau stability and aggregation. Involve in the process of tau acetylation (K281 and K274) |
[121] |
|
NBRP1 |
Degradation of BRI2 and BRI3 |
[122] |
|
FKBP51 |
Inhibition of tau clearance and increase of tau aggregation by cooperating with Hsp90. |
[123] |
|
CHIP |
Ubiquitin-mediated degradation of Aβ42 peptide, phosphorylated tau and β-secretase 1 |
[124-127] |
|
PINK1 |
Degradation of dysfunctional mitochondria via mitophagy |
[128] |
|
Presenilin-1 |
Cleaves of βCTF to produce Aβ. |
[129] |
|
Beclin-1 |
Beclin-1 mRNA and protein levels are also reduced in human aged brain, independent of AD pathology. Decreased Beclin-1 expression impairs autophagosome formation and leads to the accumulation of Aβ |
[130] |
|
CCT/TRiC |
CCT expression is reduced in aging and neurodegenerative diseases. Decreased CCT promotes LC3 degradation, improper autophagosome formation, and tau folding |
[131-133] |
Table 1 (continued). Molecular pathways of RNA and proteins related to age-related diseases
|
RNA or proteins |
Mechanisms of regulation |
References |
|
Neurodegenerative diseases-Parkinson's disease (PD) |
||
|
ATP13A2/PARK9 |
Preventing the accumulation and aggregation of α-synuclein in the dopaminergic cells |
[137] |
|
UCHL1/PARK5 |
Role as deubiquitinase or hydrolase enzymes in the UPS. |
[139-142] |
|
LRRK2 |
Mutation of LRRK2 cause the abnormal mitochondrial dynamics and the aberrant au-tophagic-lysosomal pathway, leading to cell death and α-synuclein accumulation |
[143] |
|
PINK1, Parkin |
S-nitrosylation of PINK1 and Parkin disrupt mitophagy, accumulating damaged mitochondria. |
[144] |
|
Neurodegenerative diseases-Huntington's disease (HD) |
||
|
HTT |
Modulator of selective autophagy. Required for autophagy recognition and activating machinery (p62 and ULK1). |
[145,146] |
|
Muscle disease |
|
|
|
PABPN1 |
Regulation of distal polyadenylation site and proteasomal activity |
[147,148] |
|
TTP |
Destabilize MyoD mRNA by recruiting the deadenylase components, Not1 and Caf1 |
[151,152] |
|
AUF1 |
Maintenance and differentiation of MuSCs via the regulation of mRNA stability |
[153] |
|
MuRF1 |
Regulation of MHC-related proteins to induce overall protein degradation |
[154-156] |
|
Atrogin-1 |
Degradation of eIF3-f, myogenin, and MyoD to affect protein synthesis and myogenic differentiation |
[154-156] |
|
TRAF6 |
Prevention of Beclin1, p62, and LC3B through ubiquitination |
[157,158] |
|
ATG7 |
Atg7-deficient mice induces muscle atrophy, inflammation, and muscle weakness by mitochondrial dysfunction and oxidative stress |
[160,161] |
|
Cardiovascular disease |
||
|
miR-21 |
Repression of PPARa mRNA by directly binding to 3'UTR |
[162,163] |
|
miR-126 |
Repression of proinflammatory TNF-α mRNA |
[162] |
|
ZFP36 |
Regulation of MCP-1 and IL-6 mRNA stability |
[164] |
|
NEDD4L |
Degradation of epithelial sodium channel (ENaC) through ubiquitination |
[165,166] |
|
SMURF1 |
Degradation of BMPR2 through ubiquitination |
[167] |
|
ATG5 |
Atg5-deficient mice exhibit cardiovascular diseases, including cardiac hypertrophy, contractile dysfunction, and left ventricular dilatation. Overexpression of Atg5 activates autophagy, delays cardiac aging, and extends the lifespan in mice |
[170,174] |
|
PINK1 |
Knockout of Pink1 induces cardiac hypertrophy and left ventricular dysfunction through impaired mitochondrial biogenesis in mice |
[171] |
|
Parkin |
Cardiomyocyte-specific overexpression of Parkin improves cardiac function and reduces the positivity of senescence-associated β-galactosidase (SA-β-Gal) through increasing mitophagy in aged mice |
[173] |
|
Mnf2 |
Protein required for mitochondrial dynamics and associated with mitophagy |
[172] |
Point 2-2: in order to identify the points of similarity, ii) to adjust the style. I can add as a minor observation that for the expression “Quality control” in some cases the authors used the acronym QC while in others they don’t, without a logic, but it is the only thing, for the rest, it is a well written and clear review.
Response 2-2: We apologize for this confusion. As your comment, we changed the expression “Quality control” to “QC” to adjust the style.
Reviewer 2 Report
This is a detail-rich, insight-poor review of RNA and protein processing in the context of senescence, aging, and age-related disease. Many specific examples are related where RNA or protein quality control might play a role in age-related phenomena. The difficulty in making sense of these observations is that to my knowledge there are no biological processes that do not show deficits with age. Thus, the important question is the degree to which any specific process contributes to aging or an age-related disease. This question is not addressed in the many examples described in this review. For example, while there are plausible mechanisms by which RNA or protein quality control could play a role in Alzheimer's disease, it appears none of the > 30 AD risk alleles identified by GWAS are involved in RNA or protein quality control - so is there any evidence for a causal role of quality control in AD? This review is also weakened by conflating "cell senescence" and aging. Senescent cells make up only a minority of any aging tissue, so data generated from whole tissues (e.g, an AD brain) do not report on what is going on in senescent cells. The authors would be better served if they separated observations made in senescent cells and those made in aging tissue.
Minor points:
Lines 73-76. The connection between "nuclear-retained mRNAs" and poison exon-containing mRNAs exported to the cytoplasm is unclear.
Line 101. "EJC" is used before it is defined.
Lines 238-240. All somatic cells in C. elegans are post-mitotic, so it is unclear if this model organism experiences "cell senescence". The reduction of NMD with age in worms is therefore unlikely to be relevant to cell senescence.
Line 381. Abeta generated is BACE1 is not "abnormal": Abeta 1-40 and 1-42 are routinely produced peptides by numerous cell types.
Lines 398-400. There is actually better evidence for USP11 rather than USP13 fo ar rile in Alzheimer's (Yan Y. et al, 2022).
Author Response
Response to Reviewer 2 Comments
Point 1: This is a detail-rich, insight-poor review of RNA and protein processing in the context of senescence, aging, and age-related disease. Many specific examples are related where RNA or protein quality control might play a role in age-related phenomena. 1) The difficulty in making sense of these observations is that to my knowledge there are no biological processes that do not show deficits with age. Thus, the important question is the degree to which any specific process contributes to aging or an age-related disease. This question is not addressed in the many examples described in this review. For example, while there are plausible mechanisms by which RNA or protein quality control could play a role in Alzheimer's disease, it appears none of the > 30 AD risk alleles identified by GWAS are involved in RNA or protein quality control - so is there any evidence for a causal role of quality control in AD? This review is also weakened by conflating "cell senescence" and aging. Senescent cells make up only a minority of any aging tissue, so data generated from whole tissues (e.g, an AD brain) do not report on what is going on in senescent cells. 2) The authors would be better served if they separated observations made in senescent cells and those made in aging tissue.
Point 1-1: The difficulty in making sense of these observations is that to my knowledge there are no biological processes that do not show deficits with age. Thus, the important question is the degree to which any specific process contributes to aging or an age-related disease. This question is not addressed in the many examples described in this review. For example, while there are plausible mechanisms by which RNA or protein quality control could play a role in Alzheimer's disease, it appears none of the > 30 AD risk alleles identified by GWAS are involved in RNA or protein quality control - so is there any evidence for a causal role of quality control in AD?
Response 1-1: We appreciate your valuable comment. We added the description of biological processes that deficits with age and that contributes to aging or age-related diseases as follows:
In revised manuscript:
Lines 351-353: “TRIM25 is highly expressed in HCC tissues relative to adjacent normal tissues. In colon cancer cells, TRIM25 expression is increased upon ER stress and upregulated TRIM25 reduces Keap1, an inhibitor of Nrf2, by its ubiquitination and degradation.”
Lines 378-380: “FBXW7 is inactivated by genomic deletion, genetic mutation and hypermethylated promoter in human colorectal, gastric, and breast cancers [104].”
Lines 416-418: “RNF182, a brain-enriched E3 ligase, is upregulated in tissues from AD brains. RNF182 is increased in neurons and astrocytes treated with oxygen and glucose deprivation (OGD).”
Lines 431-434: “FKBP51, which is a co-chaperone with Hsp90, is highly expressed in the aged and AD brain. FKBP51 gene is demethylated and its expression is increased with aging and AD progression. In mouse model, FKBP51 and Hsp90 blocks the clearance of tau and induces tau oligomerization, increasing tau toxicity [123].”
Lines 478-493: “UCHL1 (also known as PARK5) is a DUB enzyme and is highly expressed in neurons. In German familial PD, UCHL1 has missense mutation, an I93M substitution [139]. In cells and mouse models, UCHL1I93M mutant has highly affinity for LAMP-2A and blocks CMA-mediated degradation of α-synuclein, increasing nigral cell death [140,141]. UCHL1 is a target of oxidative damage and is modified by oxidation in aging. Oxidized UCHL1 interact with LAMP-2A, Hsp90 and Hsc70, and blocks CMA dependent degradation of α-synuclein in cells [140,142]. Mutations of LRRK2 are closely correlated with PD. Mutations (I1371V, R1441G, Y1699C, G2019S, I2012T and I2020T) in Rasp of complex proteins (ROC domain), C-terminal of Roc (COR domain) and kinase domains of LRRK2 activate their kinase activities. These mutations of LRRK2 cause the abnormal mitochondrial dynamics and the aberrant autophagic-lysosomal pathway, leading to cell death and α-synuclein accumulation [143]. Nitrosative stress is increased in aging and PD. Nitrosative stress induces S-nitrosylation of PINK1 and Parkin, and these protein modifications disrupt mitophagy, accumulating damaged mitochondria. In addition, loss of function mutations in PINK1 and Parkin genes are major cause of early-onset PD [144].”
Point 1-2: The authors would be better served if they separated observations made in senescent cells and those made in aging tissue.
Response 1-2: We appreciate your comment. We described observations made in senescent cells and those made in aging tissue as follows:
In revised manuscript:
Lines 315-321: “In cancer cells, Linc-ASEN lncRNA interacts with UPF1 and DCP1A and induces p21 mRNA decay. Depletion of Linc-ASEN lncRNA or UPF1 increases the p21 mRNA level to mediate cellular senescence in cancer cells. Linc-ASEN has inverse correlation p21 expression in tumors from patient tissues and patient-derived mouse xenograft. In addition, expression of Linc-ASEN is significantly lower in tissues from aged mouse than young mouse, and its expression shows negative correlation with p21 mRNA in mouse tissues [85].”
Lines 416-421: "RNF182, a brain-enriched E3 ligase, is upregulated in tissues from AD brains. RNF182 is increased in neurons and astrocytes treated with oxygen and glucose deprivation (OGD). Upregulation of RNF182 promotes the degradation of ATP6V0C, which is a key integral protein of gap junctions and neurotransmitter release, resulting in impaired function of gap junction and increased neuronal cell death [120].”
Lines 421-427: “The deubiquitinase, USP11, accumulates in brain tissues from AD and frontotemporal lobar degeneration with τ pathology (FTLD-tau). In HeLa cells stably expressing wildtype tau, UPS11 overexpression increases stability and aggregation of tau via its DUB catalytic activity. USP11 is involved in the process of tau acetylation (K281 and K274) in cultured cells. In addition, the expression and accumulation levels of USP11 are elevated in female mice and people than in males, exhibiting the high level of tauopathy signatures and tau tangle density [121].”
Minor points:
Minor point 1: Lines 73-76. The connection between "nuclear-retained mRNAs" and poison exon-containing mRNAs exported to the cytoplasm is unclear.
Response 1: We apologize for this confusion. As your comment, we revised the sentence as follows: “Retained introns (RIs) have two types, such as ‘poison-cassette’ exon and ‘detained’ introns”.
Minor point 2: Line 101. "EJC" is used before it is defined.
Response 2: We appreciate your comment. We added the definition of EJC as follows: “The exon junction complex (EJC) is one of protein complexes which plays a crucial role in post-transcriptional regulation for gene expression. The EJCs serve to mark exon-exon junctions after the intron removal and are displaced by ribosomes during translation [25]”
Minor point 3: Lines 238-240. All somatic cells in C. elegans are post-mitotic, so it is unclear if this model organism experiences "cell senescence". The reduction of NMD with age in worms is therefore unlikely to be relevant to cell senescence.
Response 3: As your comment, we revised the sentence as follows: “Several studies have reported that NMD activity is closely associated with aging: It is reduced during aging processes, resulting in the accumulation of abnormal RNAs due to failure of RNA QC.”
Minor point 4: Line 381. Abeta generated is BACE1 is not "abnormal": Abeta 1-40 and 1-42 are routinely produced peptides by numerous cell types.
Response 4: Thank you for your helpful comment. We revised the sentences as follows: “β-secretase 1 encoded by BACE1 cleaves APP and induces the formation of abnormal Aβ proteins and aggregates [119]”. “Mutations of Presenilin-1, which are found in the familiar AD, increase the ratio of Aβ42 to Aβ40. Increased generation of Aβ42 relative to Aβ40 is a major component of amyloid plaques in the AD brains [129]”.
Minor point 5: Lines 398-400. There is actually better evidence for USP11 rather than USP13 fo ar rile in Alzheimer's (Yan Y. et al, 2022).
Response: We appreciate your comment. We changed an example “USP13” to “USP11 (Yan Y. et al, 2022)” in the description of Alzheimer’s disease.